# Impact of Invasive Fungal Diseases on Survival under Veno-Venous Extracorporeal Membrane Oxygenation for ARDS

**DOI:** 10.3390/jcm11071940

**Published:** 2022-03-31

**Authors:** Jens Martin Poth, Jens-Christian Schewe, Christian Putensen, Stefan Felix Ehrentraut

**Affiliations:** Department of Anesthesiology and Intensive Care Medicine, University Hospital Bonn, 53127 Bonn, Germany; jens.poth@ukbonn.de (J.M.P.); jens-christian.schewe@ukbonn.de (J.-C.S.); christian.putensen@ukbonn.de (C.P.)

**Keywords:** invasive fungal disease (IFD), invasive fungal infection, extracorporeal membrane oxygenation (ECMO), acute respiratory distress syndrome (ARDS), candidiasis, candidemia, aspergillosis, bloodstream infection (BSI)

## Abstract

Objective: To assess the incidence and significance of invasive fungal diseases (IFD) during veno-venous (VV) ECMO support for acute respiratory distress syndrome (ARDS). Methods: Retrospective analysis from January 2013 to April 2021 of all ECMO cases for ARDS at a German University Hospital. In patients with IFD (IFD patients), type of IFD, time of IFD, choice of antifungal agent, duration, and success of therapy were investigated. For comparison, patients without IFD (non-IFD patients) were selected by propensity score matching using treatment-independent variables (age, gender, height, weight, and the Sequential Organ Failure Assessment (SOFA) score at ICU admission). Demographics, hospital and ICU length of stay, duration of ECMO therapy, days on mechanical ventilation, prognostic scores (Charlson Comorbidity Index (CCI), Therapeutic Intervention Scoring System (TISS), and length of survival were assessed. Results: A total of 646 patients received ECMO, 368 patients received VV ECMO. The incidence of IFD on VV ECMO was 5.98%, with 5.43% for *Candida* bloodstream infections (CBSI) and 0.54% for invasive aspergillosis (IA). In IFD patients, in-hospital mortality was 81.8% versus 40.9% in non-IFD patients. The hazard ratio for death was 2.5 (CI 1.1–5.4; *p*: 0.023) with IFD. Conclusions: In patients on VV ECMO for ARDS, about one in 17 contracts an IFD, with a detrimental impact on prognosis. Further studies are needed to address challenges in the diagnosis and treatment of IFD in this population.

## 1. Introduction

Extra-corporeal membrane oxygenation (ECMO) is increasingly used as rescue therapy in patients with acute respiratory distress syndrome (ARDS) and related indications [1,2,3,4]. In these patients, current trials and analyses suggest a 10–15% survival benefit with ECMO compared to standard therapy [3,5,6,7].

ECMO carries a high risk of severe and potentially life-threatening complications, such as complications associated with vascular access itself (i.e., bleeding, thrombosis, limb ischemia), hemorrhage due to coagulopathy, and nosocomial infections [4,8,9,10,11,12,13,14]. Bloodstream infections during ECMO are a particularly difficult problem: It has been demonstrated that central line-associated bloodstream infections (CLABSI) are associated with a significantly increased risk of death, warranting the removal of central lines whenever CLABSI is suspected [15]. Bloodstream infections on ECMO are also associated with increased mortality in pediatric patients, which is notable as removing or exchanging ECMO cannulas is not feasible [12,16]. In adults, the prevalence of nosocomial infections is >50% after >14 days of ECMO support (any site of infection) [12]. Whether these infections reduce the chance of survival remains unclear: Several studies could not find a significant impact on mortality [11,12,13,16].

When a nosocomial infection is detected in adults during prolonged ECMO therapy (>14 days), *Candida* is the most commonly found pathogen [12]. While data from the Extracorporeal Life Support Organization (ELSO) registry show that Aspergillus involvement and candidemia (*Candida* bloodstream infection, CBSI) are associated with higher mortality on ECMO, more detailed studies are lacking [17].

Here, we present the results of a single-center, retrospective study on the prevalence and the prognostic significance of invasive fungal infections (IFD) in patients undergoing veno-venous EMCO (VV ECMO) support.

## 2. Materials and Methods

We reviewed all ECMO (*n* = 646) patients from January 1st, 2013 to April 30th, 2021 from our institutional database. The analysis was approved by the local Ethics Committee and the need for individual informed consent was waived (Bonn Medical Faculty Ethics Committee #492/20).

Only patients with VV ECMO support were included in the study (Figure 1). Data on demographics, immunosuppression, and comorbidities indicated by the Charlson Comorbidity Index (CCI) [18], the status of organ failure at the time of ECMO initiation indicated by the Sequential Organ Failure Assessment Score (SOFA) [19], and length of mechanical ventilation prior to ECMO initiation were collected. The incidence of surgery or continuous renal replacement therapy (CRRT) before diagnosis of IFD was assessed. Length of stay in hospital and in the intensive care unit (ICU) was also recorded. Overall duration of ECMO support therapy, the success of ECMO weaning, general disease severity assessed by the Simplified Acute Physiology Score II (SAPS) [20], total days of mechanical ventilation, and length of survival after hospital discharge were assessed. Nursing workload was assessed by the modified Therapeutic Intervention Scoring System (TISS-10). It consists of the ten most complex procedures included in the TISS-28 [21,22]: Mechanical ventilation, multiple therapy with inotropes and vasopressors, infusion of more than five liters of fluids per day, arterial catheter, invasive hemodynamic monitoring (e.g., pulmonary artery catheter), renal replacement therapy, monitoring of intracranial pressure, treatment of metabolic acidosis or alkalosis, interventions on ICU, interventions outside of the ICU. Both scores, TISS-10 and TISS-28, correlate well [21]. Immunosuppression was defined as severe neutropenia, immunosuppressive therapy for autoimmune diseases or solid organ transplantation, acquired immune deficiency syndrome (AIDS), and immunosuppression due to oncological malignancies [23]. All values are presented as the median ± interquartile range (IQR) or mean ± standard error (SE), where applicable.

For each individual patient of the entire VV ECMO cohort, microbiological samples and tests performed during ICU stay were screened: Blood cultures (BCs) were routinely taken with every insertion of central or arterial lines. At initiation of ECMO support, bronchoalveolar lavage fluid (BALF) was collected for culture and testing. When taken, BALF was tested for galactomannan (GM), supplemented by GM testing of blood. Further samples were taken at the discretion of the treating physician. During the study period, testing for beta-D-glucan (BDG) was not available at our institution.

For this study, cultural findings of *Candida* spp. In tracheal aspirates or BALF, as well as in urine were considered to represent colonization rather than infection. All other findings of *Candidae* were considered as IFD, in agreement with the most recent definition for proven IFD [24,25]. In the case of aspergillus spp., all cultural findings in tracheal aspirates, BALF, or other body fluids and biopsies were considered IFD. Susceptibility of fungal pathogens to antifungal therapeutics was determined according to the European Committee on Antimicrobial Susceptibility Testing (EUCAST).

During the duration of the study, prophylaxis and pre-emptive therapy for candidiasis were not being used. Following current guidelines, patients with symptoms of invasive *Candida* infection and specific risk factors were treated empirically, and targeted therapy was performed in proven invasive candidiasis (positive BC) [24]. Treatment for aspergillosis was initiated in the case of positive galactomannan-testing at the discretion of the treating physician. Any case of a positive culture of aspergillosis from BALF also triggered antifungal treatment. We did not collect tissue biopsies from patients on ECMO support. As established predisposing host factors were generally missing, the definition of IA in our study does not meet the criteria for proven, probable, or possible invasive fungal disease according to the current consensus definition [25]. However, current guidelines recognize that critically ill patients without hematological malignancies are at high risk for pulmonary IA, with recommendations for prophylaxis [26,27]. In this population, Vandewoude et al. derived alternative criteria for the diagnosis of IA, acknowledging that established predisposing host factors do not have to be present [28,29]. In our study, we considered cases with mycological evidence and clinical features of aspergillosis on ECMO support as IFD.

We identified 22 patients with IFD, these were matched with 22 non-IFD patients using propensity score matching (PSM) using the treatment-independent variables: SOFA score at ICU admission, age, gender, height, and weight.

Survival between the groups was compared using Kaplan–Meier analysis and log-rank test for trend [30]. Independent risk factor analysis was performed using Cox Proportional Hazard analysis [31]. Levels of significance were determined by Mann–Whitney U or Kruskal–Wallis or Fisher’s Exact test, where appropriate. The distribution of data was analyzed by Shapiro–Wilks test. A *p*-value < 0.05 was considered significant. Data were analyzed using R Studio Version 1.4.1106.

## 3. Results

As a tertiary care center, we mainly treat critically ill surgical patients, but we also serve as a designated ECMO referral center to all medical and surgical patients of the greater Bonn region. Between January 1st, 2013 to April 30th, 2021, a total of 646 patients were treated with ECMO support at our center (Figure 1). In 22 of 368 (5.98%) patients on VV ECMO, an IFD was detected while on ECMO support (henceforth IFD patients). We used PSM to identify 22 of the remaining 346 individuals without IFD (henceforth non-IFD patients) for comparison.

As outlined in Table 1, the demographic parameters of IFD- and non-IFD patients were similar with no significant differences in the factors used for PSM: SOFA score at ICU admission, age, gender, height, and weight. Between the two cohorts, there were no differences in the etiology of respiratory failure (Table 1) and comorbidities (Appendix A): In the non-IFD cohort, there was one case of ARDS following chemotherapy and neutropenia. The incidence of surgery or CRRT was similar in both groups. Immunosuppression was observed in four (IFD) and six (non-IFD) patients. Between IFD- and non-IFD patients, there were significant differences in the median duration of ECMO support (non-IFD: 9.5 days (IQR 6.7, 13.00); IFD: 19.55 (IQR 13.85, 28.65); *p*: 0.001) and mean TISS-10 at hospital discharge (non-IFD: 17.00 (SD 10.73); IFD: 25.82 (10.13); *p*: 0.026) (Table 1).

Regarding other risk factors for IFD, the incidence of total parenteral nutrition and use of broad-spectrum antibiotics was assessed in the IFD cohort. Sixteen (72%) patients received total parenteral nutrition, with a median duration of nine days. Broad-spectrum antibiotics with a cumulative treatment duration greater than two weeks were administered in 15 (68%) patients of the IFD-cohort. ECMO was successfully weaned off in 14 (63.6%) non-IFD patients and in seven (31.8%) IFD patients (*p*: 0.069). Death before hospital discharge was more often observed in the IFD cohort (81.8%) than in the non-IFD cohort (40.9%; *p*: 0.013) (Table 1).

Survival rates of IFD and non-IFD patients are shown in Figure 2. Again, there was a significant difference between the IFD- and the non-IFD-cohort (*p*: 0.027, log-rank test for trend). In support of this, Cox proportional hazard analysis demonstrated a hazard ratio (HR) of 2.5 for death (IFD vs. non-IFD; CI 1.1–5.4; *p*: 0.023).

In the IFD cohort, we further analyzed the time course between initiation of ECMO support and diagnosis of IFD, the kind of IFD diagnosis, the time between diagnosis, and treatment and the antifungal treatment regimen (Table 2). The majority of identified fungi were *Candida* spp. (90.9%), IA was diagnosed in two cases (9.1%). We found positive BCs in 90.9% of cases (bloodstream infection, BSI), in 9.1% fungi were identified in BALF. All cases of BSI were caused by *Candida* spp., while aspergillus was identified in BALF (Table 2 and Appendix A). In 50.0% of cases, caspofungin was the first choice of treatment, voriconazole was chosen in 27.3% of cases. The fungal pathogen was susceptible to the initial treatment in every case, as determined by susceptibility testing (Appendix A). The primary treatment was changed in 12 patients after detection of IFD (Appendix A): In five cases with CBSI, fluconazole or voriconazole were replaced by either caspofungin or anidulafungin. In four cases, caspofungin was exchanged for fluconazole. In two patients, antifungal therapy was never initiated. In these two cases, results of microbiological testing and thus the diagnosis of IFD were first available at the time of death or even after the patient had deceased. The time interval between initiation of ECMO support and IFD was 12.0 days (median; IQR: 0.8, 18.0). In 54.5% of cases, clearance of infection, defined as negative follow-up BCs or samples from the site of primary infection, occurred after 3.5 days (median; IQR: 2.0, 4.3). The remaining ten patients never cleared their infection even under antifungal therapy and were classified as persistent IFD. The delay between sampling and validation of fungal culture and susceptibility testing was 5.0 days (median; IQR: 4.0, 6.0). The time between sampling and initiation of antifungal therapy was 1.5 days (median; IQR: 0.0, 1.5). In all cases with suspected IFD, empiric antifungal therapy was started before test results were validated (Appendix A, “time (d) between IFDval—treatment”). If clearance had occurred, antifungal therapy was continued for another 15 days (median; IQR: 14.0, 17.0). In the seven patients with IFD who could be weaned off ECMO support (31.8%, Table 2), antifungal therapy was continued for another eight days (median; IQR: 2.8, 11.8). One patient with IFD was discharged alive after ECMO weaning with ongoing antifungal treatment, but then lost to follow-up. Hence, the precise duration of antifungal therapy cannot be stated in this case. Two patients with established IA and 13 patients with CBSI could not be weaned off ECMO support, corresponding to 68.2% of all IFD patients.

## 4. Discussion

In this study, we report our single-center experience with the detection and treatment of IFD in a large cohort of VV ECMO patients. We utilized PSM to compare IFD and non-IFD patients. While there were no significant differences in baseline characteristics between cohorts before the initiation of ECMO support, IFD patients had longer ECMO support than non-IFD patients (Table 1). This suggests that IFD either prolongs the course of disease or that IFD are acquired during a prolonged course of disease. Indeed, the median time between initiation of ECMO support and diagnosis of IFD was 12 days, while patients without IFD were only 9.5 days (median) on ECMO support.

We observed an incidence of 5.98% for IFD, with a striking increase in mortality from 40.9% in matched non-IFD patients to 81.8% in IFD patients (Table 1 and Figure 2), corresponding to an independent HR of 2.5 (CI 1.1–5.4; *p*: 0.023) for death. We emphasize that our data do not support any conclusions on the cause of death.

CBSI constituted the majority of IFD cases, with an overall incidence of 5.4% and a mortality of 75% (16/20) in patients on VV ECMO (Table 2 and Appendix A). The incidence of IA was 0.5%, both patients died.

Going beyond previous analyses of registry data, we also performed a detailed analysis of antifungal treatment in IFD patients on VV ECMO: Antifungal treatment was started before test results were validated (i.e., empirically) and fungi were susceptible to the chosen agents in every case. In all patients with CBSI who cleared the infection, treatment was continued for more than 14 days, following current guidelines [24,32,33,34]. In one case of IA, treatment with voriconazole was initiated, the other patient deceased before diagnosis was established. Here, treatment with triazoles is in line with current guidelines [26,27].

The survival rate in our study is in line with other reports: Analysis of the Extracorporeal Life Support Organization (ELSO) registry demonstrated 48% overall survival to hospital discharge in a mixed study population of patients on VA and VV ECMO support [17]. Our survival data on patients with CBSI are comparable to data from smaller studies from ECMO patients [35]. In comparison to the general ICU population with IFD (43.4%), it was tremendously lower [36]. In contrast, the ELSO registry reported a higher rate of survival for CBSI patients on ECMO (35.9%) [17]. Of note, these patients were younger, potentially explaining the difference in survival rates. In comparison to the ELSO registry, we found a slightly higher incidence of CBSI (5.4% vs. 1.2%) [17], while the incidence of CBSI was similar in a cohort of Australian patients (6.2%) [37]. In the general ICU population with a length of stay of seven days or more, the incidence of CBSI was reported to be 3.3% [36]. While our study was not designed to identify specific risk factors for CBSI conductance on ECMO, our data suggest that CBSI are acquired during prolonged ECMO support, which is supported by current studies [35,37].

For IA, reported incidences range from 1.4% [17] to 7% [38]. We identified only two patients, which precluded further statistical analysis. In our study, sampling and testing for IFD were mostly performed at the discretion of the treating physician. Hence, it is conceivable that the incidence of CBSI and IA was underestimated.

Of note, most patients included in our study did not present classical risk factors for IFD, such as hematological malignancies, immunosuppressive therapy, or inherited defects of the immune system. Our data do not support any conclusion whether ECMO support itself imposes an additional risk for IFD conductance. This is suggested by some recent studies [39,40,41], whereas the ELSO registry analysis demonstrated an IFD incidence comparable to that of the general ICU population [17]. In our study, IFD were cleared in 54.5% of cases. Nonetheless, we emphasize that ECMO adds significant challenges to antifungal therapy: PK/PD of most antifungals is altered during ECMO, and therapeutic drug monitoring (TDM) should be performed whenever possible [42]. Furthermore, this is especially true in the case of additional renal failure with renal replacement therapy. All current guidelines also recommend the removal of central lines in CBSI, which is further supported by recent studies [24,34,43,44,45,46]. Since ECMO cannulas cannot usually be exchanged or removed, this recommendation is difficult to fulfill. Instead, guidelines suggest treatment with biofilm-penetrating echinocandins or amphotericin B, avoiding triazols [24,32,33,47,48,49].

While we strongly believe that our study adds important data to the significance of IFD during ECMO support, it is limited by its retrospective, single-center design, with a large cohort of ECMO patients, but still a limited case number of IFD. Identification of IFD patients on ECMO was hampered by the impossibility to take tissue biopsies and the impracticality of frequent CT-imaging of this patient population. In addition, BDG assays were not available and there was no standardized screening procedure for IFD. It is speculative whether any of these measures would have increased the observed incidence of IFD. Furthermore, our study was not designed to identify specific factors predisposing patients to ECMO for the conductance of IFD. We also did not assess factors potentially preventing IFD. One could speculate that the incidence of IFD is lower in patients on awake ECMO with a single dual-lumen cannula, as there is only one vascular access site and these patients are less sedated and potentially self-feeding [50].

To the best of our knowledge, our study is the first to demonstrate that the diagnosis of IFD represents an independent risk factor for death on VV ECMO, as determined by Cox proportional hazard analysis (HR 2.5; CI 1.1–5.4; *p*: 0.023), with a striking increase in mortality. Furthermore, our data also extend findings from previous and often limited nature of registry studies by the detailed evaluation of antifungal therapy in IFD patients: The detection of IFD had a tremendous negative impact on prognosis, even though treatment complied with current guidelines.

## 5. Conclusions

In summary, our study demonstrates that IFD have a detrimental impact on the prognosis of VV ECMO patients, with a striking decrease in survival. Treatment of these patients is challenging, as the ECMO circuit usually cannot be switched for source control in BSIs. Potentially, the ECMO circuit also has a significant impact on the dosing of antifungals. Future studies are needed to address these issues.

## Figures and Tables

**Figure 1 jcm-11-01940-f001:**
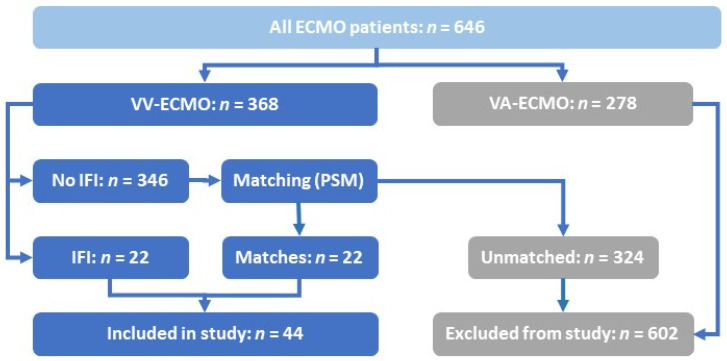
Inclusion process for the selected patient cohort. ECMO: extra-corporeal membrane oxygenation. IFD: invasive fungal diseases. PSM: propensity score matching. VA: veno-arterial. VV: veno-venous.

**Figure 2 jcm-11-01940-f002:**
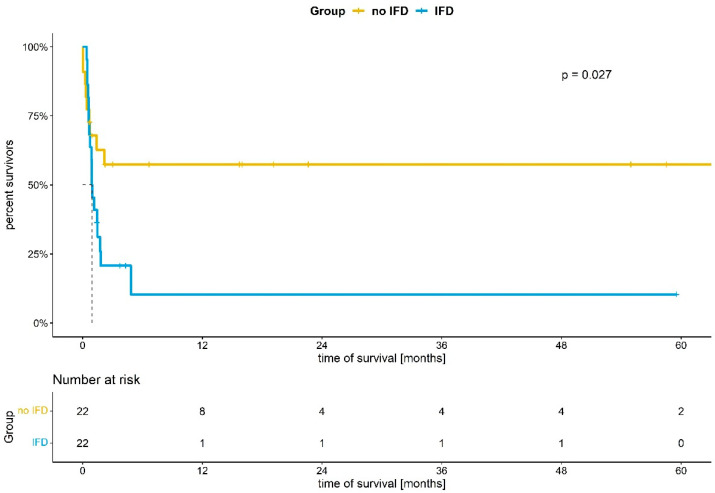
Kaplan–Meier analysis of survival in ECMO patients with and without invasive fungal disease (IFD). *p*: 0.027 (log-rank test for trend).

**Table 1 jcm-11-01940-t001:** Characteristics of patients with invasive fungal disease (IFD) and of matched patients without IFD.

Invasive Fungal Disease (*n*)	Yes (22)	No (22)	*p*
**Demographics**			
Age on ICU admission (years (mean (SD)))	57.34 (9.55)	51.6 (15.5)	0.145
Weight (kg (median [IQR]))	97.5 [80.0, 110.0]	92.5 [80.0, 121.3]	0.778
Height (cm (mean (SD)))	174.0 (7.0)	172.2 (7.4)	0.385
BMI (kg/m^2^ (median [IQR]))	31.1 [26.7, 37.7]	29.4 [27.1, 44.0]	0.897
Gender: male (*n* (%))	13 (59.1)	10 (45.5)	0.546
**ICU stats**			
Death before hospital discharge (*n*(%))	18(81.8)	9(40.9)	0.013
Days of survival (mean (SD))	124.0 (378.9)	466.6 (704.7)	0.051
Days in hospital (median [IQR])	33.1 [18.7, 50.7]	25.3 [19.3, 63.5]	0.851
Days on ICU (median [IQR])	28.1 [18.7, 48.1]	21.4 [18.0, 42.8]	0.324
ECMO duration in days (median [IQR])	19.6 [13.9, 28.7]	9.5 [6.7, 13.0]	0.001
Total days on MV support (median [IQR])	35.0 [24.3, 54.0]	23.0 [14.0, 43.8]	0.080
Days on MV prior to ECMO (median [IQR])	5.0 [0.4, 13.3]	1.0 [0.3, 2.0]	0.054
SAPS II 24hrs after ECMO initiation (mean (SD))	47.9 (11.7)	45.2 (15.3)	0.532
SAPS II at hospital discharge (mean (SD))	51.9 (13.4)	39.9 (19.7)	0.052
TISS-10 24hrs after ECMO initiation (median [IQR])	28.0 [26.0, 38.0]	27.0 [20.5, 31.0]	0.068
TISS-10 at hospital discharge (mean (SD))	25.8 (10.1)	17.0 (10.7)	0.026
SOFA Score (median [IQR])	9 [7.0, 10.0]	8.5 [7, 9.75]	0.475
**Etiology of ARDS**			
Acute Respiratory diagnosis group (%)			0.423
Aspiration pneumonitis	2 (9.1)	3 (13.6)	
Bacterial pneumonia	3 (13.6)	2 (9.1)	
Non-respiratory and chronic respiratory diagnoses	0 (0.0)	3 (13.6)	
Other acute respiratory diagnosis	6 (27.3)	6 (27.3)	
Viral pneumonia	11 (50.0)	8 (36.4)	
**Medical Procedures**			
Tracheostomy = yes (%)	8 (36.4)	11 (52.4)	0.453
CPR pre ECMO = yes (%)	0 (0.0)	5 (22.7)	0.057

BMI: body mass index. ICU: intensive care unit. ECMO: extracorporeal membrane oxygenation. SAPS II: simplified acute physiology score. TISS-10: therapeutic intervention scoring system. SOFA: sequential organ failure assessment. ARDS: acute respiratory distress syndrome. CPR: cardiopulmonary resuscitation.

**Table 2 jcm-11-01940-t002:** Diagnosis of invasive fungal disease (IFD) and treatment in patients on ECMO support.

Pathogen (*n* (%))	*C. albicans*	15 (68.2)
	*C. glabrata*	1 (4.5)
	*C. krusei*	2 (9.1)
	*C. kefyr*	1 (4.5)
	*C. parapsilosis*	1 (4.5)
	*Asp. terreus*	1 (4.5)
	*Asp. fumigatus*	1 (4.5)
Site of Infection (*n* (%))	Blood	20 (90.9)
	Lung/BALF	2 (9.1)
1st Treatment (*n* (%))	Lipo. Amphotericin B	1 (4.5)
	Caspofungin	11 (50.0)
	Fluconazole	2 (9.1)
	Voriconazole	6 (27.3)
Clearance (*n* (%))		12 (54.5)
Time (d) (Median (IQR))	ECMO-IFD	12.0 (0.8, 18.0)
	IFD validation	5.0 (4.0, 6.0)
	IFD treatment	1.5 (0.0, 1.5) *
	IFD clearance	3.5 (2.0, 4.3)
Successful Weaning of ECMO (*n* (%))		7 (31.8)

Lipo.: liposomal. *: two patients were not treated, as IFD diagnosis was established with/after death.

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
