# Peer review of "Impact of Invasive Fungal Diseases on Survival under Veno-Venous Extracorporeal Membrane Oxygenation for ARDS"

_jcm, 2022, doi:10.3390/jcm11071940_

Round 1
Reviewer 1 Report
In the Abstract, It is stated that "In IFD-patients, in-hospital mortality was 81.8% versus 40.9% in non-IFD-patients." However, in the Table 1, Death before hospital discharge was 18.2% in the IFD, 59.1% in the non-IFD. Is that correct?
Author Response
The reviewer points out a potentially misleading description in Table 1. We made the following adjustments.
In Table 1: ICU Stats
“Death before hospital discharge = no (%)” was mistaken by the reviewers a number and percent, where it was meant as indicator for the variable, i.e. “no Death in hospital before discharge occurred”. We apologize for the very unfortunate phrasing and changed it accordingly. The item is now described a “Death before hospital discharge (%)” and the numbers now match the numbers in the abstract and the manuscript’s main body (p6 l 148). We thank the reviewers for their keen observation.
Reviewer 2 Report
The authors have done a very thorough job answering my queries, though admittedly it wasn't easy to follow the corrections because in the response letter specific page/line numbers were not referenced, and I only received a marked copy of the revised manuscript, which contained so many tracked changes that it was quite disorienting (presumably some of the changes reflected the comments of other reviewers). Nonetheless, from what I could gather the authors have done a creditable job addressing my concerns, and it's unlikely that further changes would lead to substantial improvement over the current version.
Author Response
No further adjustments were required. We thank you for your time and effort.
Reviewer 3 Report
The authors have reformatted the manuscript following my comments and have answered most of my queries. The discussion section reads well and the authors need to be congratulated on this. I have a set of minor but important questions which needs clarity before the article is accepted for publication.
1. Details on parenteral nutrition and on the use of broad-spectrum antibiotics(BSAB), both risk factors for IFD: The authors have replied their institutional practice on these aspects. To rephrase my question, can the authors provide how many patients in their ECMO cohort received parenteral nutrition. Most of the patients on ECMO receive BSAB- it would be prudent to mention how many of their patients in the cohort received BSAB for more than 2 weeks. If data on the use of parenteral nutrition or the duration of BSAB is not available , this should be explicitly mentioned in the limitations along with ''Furthermore, our study was not designed to identify specific factors such as parenteral nutrition or the use of broad spectrum antibiotics, that may predispose patients on ECMO to IFD''.
2. Following Rilinger et al., immunosupression was defined as severe neutropenia, immunosuppressive therapy for autoimmune diseases or solid organ transplantation, Aquired Immune Deficiency Syndrome (AIDS) and immunosuppression due to oncological malignancies [31]----- Suggest taking this to methodology and rephrasing it as : Immunosupression was defined as severe neutropenia, immunosuppressive therapy for autoimmune diseases or solid organ transplantation, Acquired Immune Deficiency Syndrome (AIDS) and immunosuppression due to oncological malignancies [31]
3. Table 1 : Death before hospital discharge : there seems to be a discrepancy between the table numbers and what is explained in text. Please check.
Author Response
Reply to Reviewer 3
- Page 5, line 147, In Table 1: ICU Stats
“Death before hospital discharge = no (%)” was mistaken by the reviewers a number and percent, where it was meant as indicator for the variable, i.e. “no Death in hospital before discharge occurred”. We apologize for the very unfortunate phrasing and changed it accordingly. The item is now described a “Death before hospital discharge (%)” and the numbers now match the numbers in the abstract and the manuscript’s main body (p6 l 148).
- Details on parenteral nutrition and BSAB: We thank the reviewer for this remark and rephrasing it to make his/her intention clearer. We now provide the data in the results. 16 (72%) patients received total parenteral nutrition. Similarly, 15 patients received broad spectrum antibiotics for >2weeks. We included the following part in the results section (page 6, lines 152-157) “Regarding other risk factors for IFD, incidence of total parenteral nutrition and use of broad spectrum antibiotics was assessed in the IFD cohort. 16 (72%) patients received total parenteral nutrition, with a median duration of nine days. Broad spectrum antibiotics with a cumulative treatment duration greater than two weeks were administered in 15(68%) patients of the IFD-cohort.”
However, these data are limited in their applicability, since data on previous treatment outside of our ARDS centre prior to referral cannot be assessed from our records. - We moved the sentence into the methods section, as suggested by the reviewer. The phrase “Following Rilinger et al., immunosupression was defined as severe neutropenia, immunosuppressive therapy for autoimmune diseases or solid organ transplantation, Aquired Immune Deficiency Syndrome (AIDS) and immunosuppression due to oncological malignancies [31]. “ was changed to “Immunosuppression was defined as severe neutropenia, immunosuppressive therapy for autoimmune diseases or solid organ transplantation, Acquired Immune Deficiency Syndrome (AIDS) and immunosuppression due to oncological malignancies [31].” and incorporated on page 2 line 80. The sentence was deleted from the results section.
We thank all of the reviewers for their valuable comments and their effort to improve our manuscript.
This manuscript is a resubmission of an earlier submission. The following is a list of the peer review reports and author responses from that submission.
Round 1
Reviewer 1 Report
- The authors stated that ' Non-survival to hospital discharge was more often observed in the IFD-cohort (81.8%) than in the non-IFD-cohort (40.9%; p: 0.013)', however, we could not find this important information in Table 1. Please clarify.
- The following papers should be referred:
1) Rodriguez-Goncer I. Invasive pulmonary aspergillosis isassociated with adverse clinical outcomes in critically ill patients receivingveno-venous extracorporeal membrane oxygenation. Eur J Clin Microbiol Infect Dis. 2018 Jul;37(7):1251-1257.
2) Logan C. Invasive candidiasis in critical care:challenges and future directions. Intensive Care Med. 2020 Nov;46(11):2001-2014.
3) Li T. Acute Respiratory Distress SyndromeTreated With Awake Extracorporeal Membrane Oxygenation in a Patient WithCOVID-19 Pneumonia. J Cardiothorac Vasc Anesth. 2021 Aug;35(8):2467-2470.
Author Response
We also thank the reviewers for their thoughtful comments, which we found very helpful to further improve and strengthen our manuscript. Please find a list of all comments and our individual responses below:
Reviewer 1:
- Comment:
The authors stated that ' Non-survival to hospital discharge was more often observed in the IFD-cohort (81.8%) than in the non-IFD-cohort (40.9%; p: 0.013)', however, we could not find this important information in Table 1. Please clarify.
Reply: We rephrased the sentence, now including the term “death before discharge”. Table 1 was changed accordingly.
- Comment:
The following papers should be referred:
- Rodriguez-Goncer I. Invasive pulmonary aspergillosis isassociated with adverse clinical outcomes in critically ill patients receivingveno-venous extracorporeal membrane oxygenation. Eur J Clin Microbiol Infect Dis. 2018 Jul;37(7):1251-1257.
Reply: We added this important reference in the “Discussion” section on the incidence of IA. It further supports the relevance of IA for outcome on VV ECMO, with a striking decrease in survival. Since the incidence of IA in our study was low, precluding further analyses and comparisons, we could not include a more detailed discussion of this report.
2) Logan C. Invasive candidiasis in critical care:challenges and future directions. Intensive Care Med. 2020 Nov;46(11):2001-2014.
Reply: We thank the reviewer for this reference. Logan et al. elegantly discuss the challenges in the diagnosis and treatment of candidiasis in the ICU patient. They also include a paragraph on ECMO. We included this reference in the “Discussion” section of our manuscript.
3) Li T. Acute Respiratory Distress SyndromeTreated With Awake Extracorporeal Membrane Oxygenation in a Patient WithCOVID-19 Pneumonia. J Cardiothorac Vasc Anesth. 2021 Aug;35(8):2467-2470.
Reply: Although an interesting case report, we found it difficult to include this reference in our manuscript. We included it in the second last paragraph of the “Discussion” section, speculating on the role of awake ECMO with a single dual-lumen cannula for IFD. We hope this satisfies the reviewer’s request.
Reviewer 2 Report
The authors offer a commendable single-center retrospective study detailing the epidemiology, mycology, and prognosis of IFD in adult recipients of V-V ECMO. While limited by many inherent flaws, the study nonetheless offers valuable information on a topic that is largely under-explored in the literature. The authors found IFD to be associated with greater ECMO duration, which they correctly point out to not necessarily reflect causation but rather a feature of longer ECMO runs as, in general, IFD occurred in infected patients later than the overall ECMO duration of those who avoided IFD. The authors found an association between IFD and reduced survvial. I have a number of comments for the authors' consideration:
- Instead of "relevance" in abstract and P2L57, would use "significance"
- P2L54 "missing" should be changed to "lacking"
- TISS-10 is not a commonly encountered scoring system. It would help if the authors could describe it/its components in some more detail.
- When discussing the diagnostic criteria for IPA in ICU patients, the authors may also wish to consider their approach in the context of the proposed algorithm by Blot et al (PMID 22517788)
- Although the authors mention some of this in the text of the Discussion, it would help to get more information about the ICU patients up front in the Methods/Results sections, whether as text or as part of Table 1 as appropriate:
a. Was this a medical or mixed ICU--ie, any surgical patients
b. Any TPN recipients?
c. Any immunosuppressed patients, steroid recipients, etc?
d. Any information on BDG levels in the IFD group?
6. Again, the authors mention their ICU's approach to cultures/biomarkers in the Discussion section, but this would be better served earlier in the manuscript, likely the Methods section. Is there routine use of BDG, etc? Same goes for their non-use of fungal prophylaxis in ECMO--this should be revealed earlier in the manuscript rather than in the Discussion to appropriately set the stage.
7. Changes discussed in #6 above should also help shorten the Discussion section, which is quite long as it currently stands.
8. It would help to see % of patients in each group who were weaned from ECMO. Obviously all hospital survivors were weaned, and we are given survival numbers, but some non-survivors may have died on ECMO while others may have died following wean from ECMO while still in hospital.
Author Response
Reviewer 2:
The authors offer a commendable single-center retrospective study detailing the epidemiology, mycology, and prognosis of IFD in adult recipients of V-V ECMO. While limited by many inherent flaws, the study nonetheless offers valuable information on a topic that is largely under-explored in the literature. The authors found IFD to be associated with greater ECMO duration, which they correctly point out to not necessarily reflect causation but rather a feature of longer ECMO runs as, in general, IFD occurred in infected patients later than the overall ECMO duration of those who avoided IFD. The authors found an association between IFD and reduced survvial. I have a number of comments for the authors' consideration:
- Comment:
Instead of "relevance" in abstract and P2L57, would use "significance"
Reply: We made the suggested changes, and also changed the last sentence of the discussion accordingly.
- Comment:
P2L54 "missing" should be changed to "lacking"
Reply: We changed the text accordingly.
- Comment:
TISS-10 is not a commonly encountered scoring system. It would help if the authors could describe it/its components in some more detail.
Reply: We now included a brief paragraph in the “Methods” section, describing the TISS-10 and its correlation with the TISS-28.
- Comment:
When discussing the diagnostic criteria for IPA in ICU patients, the authors may also wish to consider their approach in the context of the proposed algorithm by Blot et al (PMID 22517788).
Reply: We thank the reviewer for this reference. In this study, Blot et al. externally validated the clinical algorithm for IPA diagnosis proposed by Vandewoude et al1. Both authors agree that critically ill patients are at increased risk for invasive aspergillosis. They conclude that established predisposing host factors - as described by the EORTC2 – are not required required for diagnosis in each case. We included the above mentioned studies in the description of our test strategy and definition of IA in the “Methods” section.
- Comment:
Although the authors mention some of this in the text of the Discussion, it would help to get more information about the ICU patients up front in the Methods/Results sections, whether as text or as part of Table 1 as appropriate:
- Was this a medical or mixed ICU--ie, any surgical patients
- Any TPN recipients?
- Any immunosuppressed patients, steroid recipients, etc?
- Any information on BDG levels in the IFD group?
Reply: We do agree with the reviewer that patients and treatment facility should be described in more detail. We now added a sentence at the beginning of the results section, describing our ICU further: While we deliver care for surgical patients, our ICU is a supra-regional referral center for all, i.e., medical and surgical, patients requiring ECMO support. We also added significant data on patient comorbidities, which can now be found in Table S1. This table also includes data on immunosuppression, CRRT and surgery. The table is referenced at the beginning of the “Results” section. We consider this data important to compare our study population to other patient populations.
We also do recognize TPN as an important risk factor for IFD in critically ill, non-neutropenic patients 3. At our institution, we establish enteral nutrition (e.g., feeding via a nasogastric tube) as early as possible, supplementing parenteral nutrition when needed or forced to (e.g., in the case of massive reflux, shock etc.), following current guidelines 4. However, our study was designed to evaluate the significance of IFDs for the outcome of patients on VV ECMO support. It is beyond the scope of our manuscript to determine individual risk factors for IFD. Hence, we did not include further data on the use of antibiotics and TPN, both of which might predispose patients to IFD.
Regarding the requested information on BDG levels, we kindly refer the reviewer to our reply to his comment #6.
- Comment:
Again, the authors mention their ICU's approach to cultures/biomarkers in the Discussion section, but this would be better served earlier in the manuscript, likely the Methods section. Is there routine use of BDG, etc? Same goes for their non-use of fungal prophylaxis in ECMO--this should be revealed earlier in the manuscript rather than in the Discussion to appropriately set the stage.
Reply: We agree with the reviewer and re-arranged the entire “Discussion” section accordingly. Details on our center’s approach to diagnosis and treatment of IFDs during ECMO can now be found in the “Methods” section. During the study period, testing for BDG was not available at our institution.
- Comment:
Changes discussed in #6 above should also help shorten the Discussion section, which is quite long as it currently stands.
Reply: We made significant changes to the “Discussion” section, which is now shorter and more concise (word count: 955 vs. 1693). We hope that this re-arrangement fulfills the reviewer’s request.
- Comment:
It would help to see % of patients in each group who were weaned from ECMO. Obviously all hospital survivors were weaned, and we are given survival numbers, but some non-survivors may have died on ECMO while others may have died following wean from ECMO while still in hospital.
Reply: We added the required information in the results section. After successful withdrawal of ECMO support, two non-IFD-patients died and three IFD-patients.
Reviewer 3 Report
The authors have done a propensity matched analysis on invasive fungal infections in patients needing VV ECMO. The manuscript deals with an important topic and is novel. However it needs further improvement both in terms of presentation as well as organisation of discussion. I have collated my comments below.
COMMENTS
Abstract:
Conclusion: Rather than saying IFD occurs frequently, this sentence should be rephrased as About 1 in 6 patients get IFD…….
Introduction:
Blood stream infections on EMCO are also associated with increased mortality---- please change EMCO to ECMO.
Methodology:
The authors do not mention in detail about the risk factors of fungal infection for their VV ECMo population. They make a passing mention of their patients not having hematological malignancies and immunosuppressive therapiess in the discussion section. I am sure further details can be pulled up from their database. An idea of how many patients had other forms of immunosuppression (cancer, steroids) as well as duration of antibiotic therapy will be important to ascertain the incidence of IFD in this population. The authors should also include details of diabetes mellitus, renal insufficiency, the use of broad-spectrum antibiotics, parenteral nutrition, haemodialysis, and solid organ cancer in their tables to give a holistic picture of other risky cofactors.
Discussion:
This section is very long and needs to be cut down into 4 paragraphs – 1. Summary of the findings of this study. 2. Comparison to other cohorts 3. Possible reasons and solutions and 4. Strengths and limitations.
Currently lines 184-278 discuss more about diagnosis of CBSI and various societal guidelines. This is less useful when ECMO lines cannot be changed and hence guidelines that say infected lines need to be removed would be obsolete. Also it is logical that if central lines and arterial catheters are infected, ECMO cannulas will have biofilms of these pathogens. One can only treat them with antifungals and hope things would resolve. I suggest replacing this entire segment with an idea on how biofilms are formed around the cannulas, the limitations of guidelines in this scenario and how best IFD can be treated in VV ECMO patients given the issues with pharmacokinetics of antifungals in the ECMO circuit.
Also reporting on anecdotal cases (eg: C. parapsilosis) should be avaoided and the discussion should focus on the big picture.
References : these are not arranged as per journal requirements. Please chk them
Author Response
Reviewer 3:
The authors have done a propensity matched analysis on invasive fungal infections in patients needing VV ECMO. The manuscript deals with an important topic and is novel. However, it needs further improvement both in terms of presentation as well as organisation of discussion. I have collated my comments below.
- Comment:
Abstract:
Conclusion: Rather than saying IFD occurs frequently, this sentence should be rephrased as About 1 in 6 patients get IFD…….
Reply: We rephrased the sentence accordingly, translating the observed incidence of IFD during VV ECMO (5.98%) to “one in 17”.
- Comment:
Introduction:
Blood stream infections on EMCO are also associated with increased mortality---- please change EMCO to ECMO.
Reply: We corrected the spelling error.
- Comment:
Methodology:
The authors do not mention in detail about the risk factors of fungal infection for their VV ECMo population. They make a passing mention of their patients not having hematological malignancies and immunosuppressive therapiess in the discussion section. I am sure further details can be pulled up from their database. An idea of how many patients had other forms of immunosuppression (cancer, steroids) as well as duration of antibiotic therapy will be important to ascertain the incidence of IFD in this population. The authors should also include details of diabetes mellitus, renal insufficiency, the use of broad-spectrum antibiotics, parenteral nutrition, haemodialysis, and solid organ cancer in their tables to give a holistic picture of other risky cofactors.
Reply: We do agree with the reviewer’s comment. The requested data is necessary to compare our study population to other patient populations and to test our data for plausibility. For these reasons, we added significant data to the manuscript: First, we describe our facility in more detail (surgical ICU, but also a supra-regional referral center for all, i.e., medical and surgical, patients requiring ECMO support). We also added significant data on patient comorbidities, which can now be found in Table S1. This table also includes data on immunosuppression, CRRT and surgery. The table is referenced at the beginning of the “Results” section.
The reviewer also asks for details on parenteral nutrition and on the use of broad-spectrum antibiotics, both risk factors for IFD3: At our institution, we establish enteral nutrition (e.g., feeding via a nasogastric tube) as early as possible, supplementing parenteral nutrition when needed or forced to (e.g., in the case of massive reflux, shock etc.), following current guidelines 4. Similarly, we do follow the current German guideline on the use of empirical antibiotic therapy, which is regularly utilized in critically ill patients on VV ECMO5. Our study was designed to evaluate the significance of IFDs for the outcome of patients on VV ECMO support. It is beyond the scope of our manuscript to determine all individual risk factors for IFD, which were not all recorded prospectively. Hence, we did not include further data on the use of antibiotics and TPN. However, we do strongly believe that we added significant data, fulfilling most of the reviewers requests.
- Comment:
Discussion:
This section is very long and needs to be cut down into 4 paragraphs – 1. Summary of the findings of this study. 2. Comparison to other cohorts 3. Possible reasons and solutions and 4. Strengths and limitations.
Currently lines 184-278 discuss more about diagnosis of CBSI and various societal guidelines. This is less useful when ECMO lines cannot be changed and hence guidelines that say infected lines need to be removed would be obsolete. Also it is logical that if central lines and arterial catheters are infected, ECMO cannulas will have biofilms of these pathogens. One can only treat them with antifungals and hope things would resolve. I suggest replacing this entire segment with an idea on how biofilms are formed around the cannulas, the limitations of guidelines in this scenario and how best IFD can be treated in VV ECMO patients given the issues with pharmacokinetics of antifungals in the ECMO circuit.
Also reporting on anecdotal cases (eg: C. parapsilosis) should be avaoided and the discussion should focus on the big picture.
Reply: We thank the reviewer for the helpful comment, We re-wrote and re-arranged large parts of the “Discussion” section, which now is also shorter (Now: 955 words. Previously: 1693 words). We followed the reviewer’s suggestion to structure the section, but obviously the comparison to other cohorts and the discussion of differences are closely linked and blended to some extent. We left out “anecdotal cases” and did not discuss C. parapsilosis. Also, we did not further discuss aspergillosis, as low case numbers in our study precluded further analysis. We hope that this satisfies the reviewer’s request.
- Comment:
References : these are not arranged as per journal requirements. Please chk them
Reply: We changed the references accordingly.
References:
- Vandewoude, K. H. et al. Clinical relevance of Aspergillus isolation from respiratory tract samples in critically ill patients. Crit. Care Lond. Engl. 10, R31 (2006).
- Donnelly, J. P. et al. Revision and Update of the Consensus Definitions of Invasive Fungal Disease From the European Organization for Research and Treatment of Cancer and the Mycoses Study Group Education and Research Consortium. Clin. Infect. Dis. 71, 1367–1376 (2020).
- León, C. et al. Fungal colonization and/or infection in non-neutropenic critically ill patients: results of the EPCAN observational study. Eur. J. Clin. Microbiol. Infect. Dis. Off. Publ. Eur. Soc. Clin. Microbiol. 28, 233–242 (2009).
- Elke, G. et al. [DGEM Guideline ‘Clinical Nutrition in Critical Care Medicine’ - short version]. Anasthesiologie Intensivmed. Notfallmedizin Schmerzther. AINS 54, 63–73 (2019).
- Bodmann, K.-F. & Paul-Ehrlich-Gesellschaft f??r Chemotherapie e.V. Kalkulierte parenterale Initialtherapie bakterieller Erkrankungen bei Erwachsenen - Update 2018 PEG S2k Leitlinie (AWMF-Registernummer 082-006). (2019).